# Study on the Effect of Radish Sprouts on Short-Chain Fatty Acids and Gut Microbial Diversity in Healthy Individuals

**DOI:** 10.3390/foods14020170

**Published:** 2025-01-08

**Authors:** Ru Li, Xuehong Chen, Cong Shi, Yi Zhu

**Affiliations:** 1College of Food and Biological Engineering, Xuzhou Institute of Technology, Xuzhou 221018, China; liru819902@163.com (R.L.); xzcxhong@163.com (X.C.); 15996958914@163.com (C.S.); 2College of Food Science and Nutritional Engineering, China Agricultural University, Beijing 100083, China

**Keywords:** radish sprouts, simulated gastrointestinal digestion, in vitro fermentation, short-chain fatty acids, gut microbiota

## Abstract

This study aimed to assess the impact of radish sprouts on the gut microbiota of healthy individuals. Radish sprout additives, subjected to short-term storage and steam treatment, were used to intervene in an in vitro culture of human gut microbiota. The influence of radish sprouts on the gut microbiota was evaluated by monitoring short-chain fatty acid (SCFA) content and proportion in the fermentation broth, and microbial diversity was assessed using 16S rDNA amplicon sequencing. The results indicated that the gut microbiota produced a substantial amount of SCFA within 48 h of fermentation, with a right-skewed distribution across all groups. The addition of both digestates enhanced *Firmicutes* diversity, while *Bacteroidetes* and *Proteobacteria* diversity remained stable between the control and fresh sprout groups. The 30 s steam treatment group showed an increase in *Bacteroidetes* and a decrease in *Proteobacteria* diversity. The abundance of *Bacilli*, *Bacillaceae*, and *Bacillus* was significantly higher in both the fresh and steam-treated groups compared to the control. Both fresh and steam-treated radish sprout digestates enriched gut microbiota diversity, with steam treatment showing superior effects. These findings suggest that radish sprout consumption may positively influence gut microbiota, with steam treatment potentially enhancing these benefits.

## 1. Introduction

The human microbiome, with the majority residing in the gut, is composed of an immense number of symbiotic microorganisms, particularly in the colon where counts reach 10^11^ to 10^12^ cfu/mL [1,2]. This gut microbiota, boasting potent physiological functions, is considered an essential microecosystem for human health [3]. Nevertheless, the equilibrium of the gut microbiota is highly susceptible to modulation by diverse factors. Disruptions in this balance have been implicated in a spectrum of diseases, with connections now established between gut microbiota dysbiosis and conditions affecting both the gastrointestinal tract and the brain [4].

Diet has been established as a significant determinant of gut microbiota composition and metabolic activities [5]. Research, both in vitro and in vivo, demonstrates that dietary polyphenols from fruits and vegetables can selectively suppress harmful gut bacteria while fostering the growth of beneficial species such as Bifidobacterium [6,7]. This balance is further maintained by reducing the prevalence of detrimental microbes like Bacteroides, Enterococcus, and Staphylococcus [8,9]. Concurrently, the gut microbiota engages in the metabolism of dietary components, influencing their health functionalities [10]. Extensive research underscores the profound impact of the interplay between gut microbiota and dietary constituents on human physiology [11]. Deciphering the intricate relationships and mechanisms among dietary components, gut microbiota, and physiological health is not only of paramount importance but also a focal point of contemporary scientific inquiry.

Radish sprouts (*Raphanus sativus* L.), germinated from radish seeds, are enriched with carbohydrates, an array of vitamins, dietary fiber, and amylase [12]. These nutrients promote the absorption of nutrients, decrease the consumption of additional foods, and stimulate intestinal motility, assisting in digestion and easing constipation, which can aid in weight management [13]. The sprouts also contain bioactive compounds, including phenolics, which exhibit antioxidant activities and are hypothesized to influence the gut microbiota’s structure and function [14]. Phenolic compounds in radish sprouts, such as phenolic acids and flavonoids, are known for their antioxidant and anti-inflammatory properties. Specific phenolics like sinapic acid, ferulic acid, and kaempferol glycosides have been identified in radish sprouts [15]. These compounds are suggested to contribute to the prebiotic effect on the gut microbiota by selectively promoting the growth of beneficial bacteria, such as Bifidobacterium and Lactobacillus species, while inhibiting the growth of potentially harmful bacteria [7]. Beyond their prebiotic potential, these sprouts are low in calories and high in nutrients, making them a valuable component of a balanced diet. They are particularly rich in vitamin C, which supports the immune system, and vitamin K, which is essential for blood clotting and bone health [13].

Short-chain fatty acids (SCFAs) are predominantly generated as metabolic byproducts following the fermentation of undigested dietary substances, such as fiber from fruits and vegetables, by microorganisms within the large intestine [16]. Specific gut microbiota, including beneficial species within the Bifidobacterium genus, contribute to SCFA production by fermenting carbohydrates and proteins present in the intestinal chyme [17]. Research indicates that carotenoids present in radish sprouts are intricately linked to the gut microbiota [18]. These carotenoids, including β-carotene, lutein, lycopene, and astaxanthin, have been shown to modulate the gut microbiota’s composition by enhancing the growth of beneficial bacteria and suppressing the proliferation of harmful species [19]. Notably, these carotenoids significantly augment the overall production of SCFA and enrich the population of beneficial bacteria like *Roseburia* and *Parasutterella*, while concurrently curbing the growth of detrimental bacteria such as *Collinsella* [20,21]. The polyphenolic content in radish sprouts is also linked to the enhancement of short-chain fatty acid (SCFA) production, which plays a crucial role in maintaining gut health and overall well-being [16].

In recent years, the development of in vitro gut microbiota fermentation technology has greatly facilitated the study of gut microbiota, allowing for a thorough investigation into the effects of additives on the microbiota and its metabolites [22]. Therefore, based on the bioactive substances present in radish sprouts, fecal samples are collected from individuals who meet the requirements according to health and dietary surveys. The impact of stored and steamed radish sprout digestates on human gut microbiota is studied through in vitro culture, with the changes in the content and proportion of SCFA in the fermentation broth reflecting the influence of the functional substances in radish sprouts on the gut microbiota. Concurrently, high-throughput sequencing methods (16S RNA sequencing) are employed to investigate the diversity of the gut microbiota in the fermentation broth, thereby exploring the effects of radish sprout digestates on the gut microbiota. The novelty of our study lies in its comprehensive approach to understanding the impact of radish sprouts on the gut microbiota. While previous studies have explored the effects of various sprouts on gut health, our research delves specifically into the role of radish sprouts in modulating microbial diversity and function. By employing in vitro fermentation models and 16S rDNA sequencing, we provide a deeper insight into the mechanisms by which radish sprouts may exert their prebiotic effects.

## 2. Materials and Methods

### 2.1. Radish Sprout Cultivation

Plump, mold-free Shengfeng white radish seeds (from Beijing Jingyan Shengfeng Seed Research Institute) were selected. The radish seeds were thoroughly cleaned and then soaked in distilled water for 5 h. After soaking, the seeds were rinsed 2–3 times. The soaked seeds were evenly spread on a 30 × 20 cm seedling tray lined with four layers of gauze, with approximately 200 seeds per tray. The seeds were germinated in the dark at 25 °C for 3 days, with deionized water sprayed in the morning and evening. After three days, light was reintroduced with an intensity of 100 mmol/m^2^/s, under a 16 h/8 h (day/night) photoperiod, at a temperature of 25 °C and a relative humidity of 70% to 80%. After 7 days, the radish sprouts were ready for harvest [15].

### 2.2. Acquisition of Additives for Radish Sprout Cultivation

Uniformly cultivated radish sprouts were subjected to short-term storage (at 25 °C for 12 h (S1) and at 4 °C for 12 h (S2)) and short-term steaming (in boiling water for 30 s (P1) and 300 s (P2)). These treated sprouts, along with fresh radish sprouts (fresh sprouts, FS), were ground into powder with liquid nitrogen and stored at −80 °C for subsequent in vitro simulated gastrointestinal digestion [15,23]. **Simulated saliva** was prepared by dissolving 2.38 g of Na_2_HPO_4_, 0.19 g of KH_2_PO_4_, and 8 g of NaCl in 1 L of deionized water, with the pH adjusted to 6.75, and the addition of 12.5 mg of α-amylase (E.C. 3.2.1.1, A3176-500KU, 16 U/mg solid, Sigma, Kawasaki-shi, Japan) to achieve an enzyme activity of 200 U/mL. **Simulated gastric fluid** was prepared by adding pepsin A (EC3.4.23.1, P7012-250 MG, 3641 U/mg protein, 87% protein content, Sigma) (from porcine gastric mucosa) to 0.03 mol/L NaCl to achieve an enzyme activity of 300 U/mL, with the pH adjusted to 1.2. **Simulated intestinal fluid** was prepared by dissolving 0.05 g of pancreatin (equivalent to 4 times USP, P1750-25G, Sigma) and 0.3 g of bile extract (B8631-100g, Sigma) in 35 mL of 0.1 mol/L NaHCO_3_. **The final solution** contained 120 mmol/L NaCl and 5 mmol/L KCl. **For the in vitro simulated gastrointestinal digestion process**, 5 g of each sample powder was homogenized in 5 mL of simulated saliva and incubated at 37 °C for 10 min. Then, the samples were adjusted to pH 1.2 with HCl (5 mmol/L), suspended in 15 mL of simulated gastric fluid, and incubated at 37 °C for 120 min. Subsequently, the samples were adjusted to pH 6 with 0.1 mol/L NaHCO_3_, suspended in 15 mL of simulated intestinal fluid, adjusted to pH 7 with 1 mol/L NaOH, and then 5 mL of the final solution was added, with in vitro digestion for 120 min. The final concentration of each sample was 0.1 g/mL [24,25].

### 2.3. Assessment of Health and Dietary Practices in a Study Population

A survey questionnaire was meticulously designed to evaluate the health and dietary practices of the study population, and informed consent was secured from all participants. Participants were excluded based on three primary criteria: the intake of antibiotics or probiotic medications within the three-month period preceding the study, doubts concerning the participants’ willingness or capacity to adhere to the experimental protocol, and the existence of significant medical conditions, including but not limited to kidney diseases requiring dialysis. A cohort of 12 participants was selected, exhibiting similar demographic profiles and dietary behaviors, such as age, gender, smoking status, alcohol consumption, and dietary habits [26].

### 2.4. Collection and Pre-Treatment of Fecal Samples

Researchers provided fecal sampling tubes, gloves, and sample information strips to the subjects. Fecal samples were collected by eligible subjects at home using the provided fecal sampling tubes. The requirements for the samples are as follows [27]: fresh, not mixed with urine, free from disinfectants and sewage, and ideally not in contact with any impurities. After sample collection, the sample information strip (including name, gender, sampling location, and sampling time (accurate to minutes)) should be filled out accurately and transferred to the laboratory’s cold storage within 3 h, and the preliminary treatment of the fecal samples should be completed within 8 h. The remaining samples are stored at −80 °C. This study has been approved by the Human Research Ethics Committee of China Agricultural University (CAUHR-2019007).

Fresh feces of 0.8 g were weighed and added to 8 mL of sterilized PBS solution (pH 7.2–7.4) (mass concentration of 0.1 g/mL), mixed evenly by shaking, and then centrifuged at 3000× *g* at room temperature for 10 min. The supernatant after sedimentation was taken for subsequent experiments [28].

### 2.5. Media for In Vitro Cultivation of Intestinal Microbiota

The medium consists of (per 100 mL) 0.1 g of casein peptone, 0.25 g of yeast extract, 0.4 g of NaHCO_3_, 0.1 g of cysteine, 0.045 g of K_2_HPO_4_, 0.045 g of KH_2_PO_4_, 0.09 g of NaCl, 0.009 g of MgSO_4_·7H_2_O, 0.009 g of CaCl_2_, 0.1 mg of gentian violet, 1 mg of hemin, 1 mg of biotin, 1 mg of cobalamin, 3 mg of p-aminobenzoic acid, 5 mg of folic acid, and 15 mg of pyridoxine, with a pH of 6.5 [29]. The medium is sterilized by autoclaving and then distributed into 10 mL glass medical sealable vials under anaerobic conditions, 5 mL per vial, for storage at room temperature.

### 2.6. In Vitro Cultivation of Intestinal Microbiota with Radish Sprout Digesta

The experimental setup was divided into six groups, with each group containing 12 bottles, and the culture medium was appropriately marked. Alongside the FS, S1, S2, P1, and P2 groups, a control group (CK), which received no intervention substances, was also included. The operations were carried out as follows: To each bottle within the groups, 500 μL of fecal supernatant from 12 distinct subjects was introduced. Subsequently, 100 μL of the intervention substances were added, with the CK group receiving 100 μL of sterilized PBS, and the remaining groups receiving 100 μL of the respective radish sprout digesta. The cultures were incubated at 37 °C for a duration of 48 h to facilitate ex vivo fermentation. Upon completion of the fermentation period, the fermented broth was transferred to 2 mL centrifuge tubes for further processing, while the surplus samples were cryopreserved at −80 °C [30,31].

### 2.7. Detection of Short-Chain Fatty Acids

External standard compounds included acetic acid, propionic acid, butyric acid, isobutyric acid, pentanoic acid, and isovaleric acid, with crotonic acid serving as the internal standard. Standards, fermentation broths (CK, FS, S1, S2, P1, and P2), and fecal suspensions (FeS), each 500 μL in volume, were aliquoted into 1.5 mL centrifuge tubes. To each, 100 μL of a crotonic acid and metaphosphoric acid mixture was added, with a sample-to-internal standard volume ratio of 5:1. The mixtures were vortexed to achieve uniformity and then placed in a −20 °C freezer overnight. After thawing, the samples were centrifuged at 14,000 rpm for 5 min. The supernatant was collected in its entirety using a 1 mL syringe, filtered through a hydrophilic filter membrane, and transferred to fresh centrifuge tubes. A 100 μL aliquot of the filtrate was then pipetted into a sample vial equipped with a liner, ensuring that any air bubbles at the bottom of the liner were expelled prior to analysis. The gas chromatographic analysis was conducted using a GC9720 (Fuli) instrument, equipped with an HP-FFAP column (30 m × 0.25 mm × 0.25 μm, Agilent, Santa Clara, CA, USA). The injector temperature was maintained at 250 °C; the column temperature program initiated at 75 °C, with a ramp rate of 20 °C/min, and a final temperature of 220 °C for a duration of 2 min; the FID detector temperature was set at 250 °C; the gas flow rates were configured to 40 mL/min for hydrogen, 400 mL/min for air, and 20 mL/min for tail-blow nitrogen; the injection mode was split, with a split ratio of 5:1; and the injection volume was 1 μL [31,32].

### 2.8. Detection of Intestinal Microbiota Diversity

Following the outcomes of the initial experiments, fermentation broths from the CK, FS, and P1 groups were selected for further analysis. The notation F (Female) and M (Male) signifies the gender of the subjects, with the numerals indicating their respective identification numbers. For instance, CK.F.1 denotes the fermentation broth sample associated with Female subject number 1 in the control group that received no intervention substances.

The DNA extraction from fecal samples obtained post in vitro fermentation was conducted utilizing the OMEGA Stool DNA Kit (D4015). Agarose gel electrophoresis was employed to assess the purity and concentration of the extracted DNA. Aliquots of the sample DNA were placed into centrifuge tubes and diluted with sterile water to a concentration of 1 ng/μL. Subsequently, the diluted genomic DNA served as a template for PCR amplification, which was carried out using barcoded specific primers (515F and 806R) corresponding to the V3–V4 region, Phusion^®^ High-Fidelity PCR Master Mix with GC Buffer from New England Biolabs, and high-fidelity enzymes to ensure amplification efficiency and accuracy. The PCR products were electrophoresed on a 2% agarose gel to verify the integrity and were then mixed in equal amounts. The PCR products were purified using a 1×TAE buffered 2% agarose gel, and the target bands were excised and recovered. The Thermo Scientific GeneJET Gel Recovery Kit (Thermo Fisher Scientific, Waltham, MA, USA) was utilized for the purification of the excised products. Library construction was performed using the Thermofisher Ion Plus Fragment Library Kit 48 rxns (Thermo Fisher Scientific, Waltham, MA, USA), and following quantification with Qubit and library validation, the constructed libraries were sequenced on the Thermofisher Ion S5TMXL platform (Thermo Fisher Scientific, Waltham, MA, USA). Data obtained were subsequently processed for analysis.

The 16S rDNA amplicon sequencing approach was employed, targeting the hypervariable V3–V4 regions, to amplify and sequence the DNA extracts from the in vitro fermentation broths [33]. Universal primers designed for the conserved regions facilitated PCR amplification. Following amplification, the variable regions were sequenced and analyzed for microbial identification. Characteristic of the amplified region, the IonS5™ XL sequencing platform was utilized, employing single-end sequencing to construct small fragment libraries. After sequencing, the reads were trimmed and filtered (Raw data were extracted from the IonS5™ XL platform in fastq file format. Sample data were delineated based on barcode sequences, and chimeras are eliminated to obtain clean reads. Quality control metrics, including raw reads, clean reads, total base count, average length, Q20 percentage, GC content, and effective percentage, were assessed). Operational Taxonomic Units (OTUs) were clustered, and species annotation and abundance analysis were conducted to reveal the composition of species within the samples. Further complexity analysis of the samples was performed, including alpha diversity (α-diversity analysis) for intra-sample diversity assessment and beta diversity (β-diversity analysis) for inter-sample comparison to uncover differences among the samples.

### 2.9. Data Processing

Data statistics and single-factor analysis of variance (ANOVA) were conducted using Excel 2010 and SPSS 20.0 software, followed by Duncan’s multiple range test for significance analysis (*p* < 0.05). Graphs were created using Origin 9.1 software. Results are presented as mean ± standard deviation. All experiments were repeated three times.

## 3. Results and Discussion

### 3.1. The Effect of Radish Sprout Digesta Supplementation on Short-Chain Fatty Acids

Short-chain fatty acids (SCFAs), primarily consisting of acetic acid, propionic acid, and butyric acid, maintain the intestinal barrier, promote gut health, regulate blood sugar and body weight, enhance cardiac and skeletal health, possess anti-inflammatory and immune-modulating effects, and may even contribute to cancer prevention [34,35,36]. SCFAs can also improve neurological health, exerting positive effects on mood and cognitive functions [37]. The levels of acetic, propionic, and isobutyric acids in all experimental groups were elevated after 48 h of fermentation compared to the FeS group, with the exception of valeric acid (Figure 1). Post in vitro fermentation, a greater dispersion was observed in other SCFAs, and a notable right skewness was apparent, with the exception of propionic acid. The distribution of acetic acid did not yield any directional outcomes except for an increased dispersion in the S1 group. The incorporation of foreign materials in the FS, S1, S2, P1, and P2 groups resulted in higher propionic acid levels overall when compared to the control group, with more significant effects noted in the S1, S2, P1, and P2 groups, and a reduction in dispersion observed in the FS, P1, and P2 groups. The distribution of butyric acid revealed greater dispersion in the S1 group relative to CK, while S2 and P2 exhibited the opposite trend. The addition of digesta to the fermentation fluids increased isobutyric acid content, with P1 showing a less pronounced increase compared to P2, and except for increased dispersion in the P2 group, the other groups remained relatively consistent. The fermentation fluid in the P1 group demonstrated reduced dispersion of valeric acid, and the P2 group had an increased valeric acid content relative to CK. The distribution of isovaleric acid saw an increase in the S1, S2, P1, and P2 groups, with S2 and P2 displaying greater dispersion. The elevated levels of acetic, propionic, and isobutyric acids post-fermentation suggest a shift in microbial metabolism, which could be attributed to the fermentative capabilities of the introduced digestates [38,39]. Collectively, the supplementation of various digesta positively influenced the elevation in certain SCFAs in the fermentation fluid, particularly after the radish sprout digesta had been subjected to storage and steaming processes.

### 3.2. The Effect of Radish Sprout Digesta Supplementation on Gut Microbiota Diversity—A Comparative Analysis of Microbial Diversity Based on OUT

In light of initial research outcomes, the aim was to delve deeper into the effects of radish sprout digesta on intestinal microbiota. Consequently, ex vivo fermentation broths from the CK, the FS, and the P1 were chosen for further investigation. A 16S bioinformatics analysis was performed on these selected samples. The effective data from all samples were clustered into Operational Taxonomic Units (OTUs) at a 97% similarity level, after which the representative sequences of these OTUs underwent species annotation in preparation for subsequent analyses. As depicted in the bar chart illustrating the top 10 phyla by relative abundance (Figure 2), based on species annotation results, Proteobacteria emerges as the predominant phylum across all experimental groups, with *Escherichia* and *Desulfovibrio* being its principal constituents [40]. The supplementation of fresh radish sprout digesta (FS) and steamed radish sprout digesta for 30 s (P1) has been observed to modulate the relative abundance of *Proteobacteria*, with a decrease in the FS group and an increase in the P1 group. Bacteroidetes, which includes *Bacteroides* and *Prevotella* as key genera [41], ranks second in abundance and exhibits an inverse response to *Proteobacteria*, with an increase in the FS group and a decrease in the P1 group, mirroring the pattern observed for *Fusobacteria*, the fourth-ranked phylum. *Firmicutes*, the third-most abundant phylum, encompasses genera such as *Clostridium*, *Lactobacillus*, *Ruminococcus*, *Eubacterium*, *Faecalibacterium*, and *Roseburia* [42], and its relative abundance is notably higher in the FS group compared to the P1 group. *Actinobacteria*, also present in the fermentation liquids, is characterized primarily by the genus *Bifidobacterium*, with several species recognized for their probiotic properties [43].

Figure 2B illustrates a cumulative representation of the top 30 genera based on their relative abundance, categorized according to species annotation outcomes. Excluding the “other” category, *Bacteroides* (from the Bacteroidetes phylum) emerges as the predominant genus, with its abundance notably elevated in the fermentation group supplemented with fresh radish sprout digestate (FS) and diminished in the group with steamed radish sprout digestate for 30 s (P1). It has been documented that the prevalence of *Bacteroides* in individuals with type 2 diabetes significantly surpasses that in healthy populations [44]. *Fusobacterium* (belonging to the *Fusobacteria* phylum) follows suit, with its abundance influenced by the additives in a manner congruent with *Bacteroides*, showing an increase in the FS group and a decrease in the P1 group. Evidence suggests that *Fusobacterium* is more abundant in colorectal cancer patients compared to healthy individuals [45]. *Citrobacter* (*Proteobacteria phylum*) and *Parabacteroides* (*Bacteroidetes phylum*) exhibit similar proportions, while *Phascolarctobacterium* (*Firmicutes phylum*) and *Dialister* (*Firmicutes phylum*) are detected, albeit in lower quantities. *Citrobacter*, a common non-pathogenic gut bacterium that can become pathogenic under compromised immunity, bears a striking resemblance to the significant pathogen Salmonella within the gut [46]. *Parabacteroides*, integral to the microbiota of healthy individuals, is known for its production of beneficial metabolites such as acetic acid and succinic acid [47]. Research has indicated that individuals predisposed to weight loss harbor higher levels of *Phascolarctobacterium*, whereas those resistant to weight loss exhibit increased *Dialister* levels [48]. Collectively, it appears that the consumption of steamed radish sprouts for 30 s, as opposed to fresh radish sprouts, leads to a reduction in the abundance of certain bacterial genera detrimental to health, including *Bacteroides* and *Fusobacterium*. The increased diversity of *Firmicutes* and the reduction in potentially harmful genera such as *Bacteroides* and *Fusobacterium* upon steaming radish sprouts for 30 s underscore the importance of food processing methods in shaping the gut microbiota [8]. These findings have implications for the development of dietary strategies aimed at enhancing gut health and potentially mitigating the risk of associated diseases.

Species annotation and abundance data at the genus level across all samples (grouped) were analyzed using the maximum value sorting method to identify the top 35 genera (with group abundance defined as the average abundance across all samples within the group). Clustering at the species level was performed based on the abundance information in each sample, and heatmaps were generated using the pheatmap package in R 3.6.0 software. These visualizations facilitate the identification of species that are either abundant or scarce in specific samples. Figure 3 depicts the clustering heatmaps at the genus level (grouped), with values representing Z-scores derived from the standardized relative abundance of each species. The Z-score for a sample in a given taxon is calculated by subtracting the average relative abundance across all samples from the sample’s relative abundance in that taxon, and then dividing it by the standard deviation of all samples in that taxon. This approach revealed 35 of the most abundant bacterial genera, marked with different colors across the three groups. Notably, significant clustering differences were observed among the CK, FS, and P1 groups. Genera such as *Sutterella* (*Proteobacteria*), *Flavonifractor* (*Firmicutes*), *Intestinimonas* (*Firmicutes*), *Raoultella* (*Proteobacteria*), *Dorea* (*Firmicutes*), *Faecalibacterium* (*Firmicutes*), *Parabacteroides* (*Bacteroidetes*), *Holdemania*, *Streptococcus* (*Firmicutes*), *Erysipelatoclostridium*, and *Alistipes* were found to be significantly more abundant in the control group compared to the FS and P1 groups. *Sutterella* has been identified in the gut of children with autism and is associated with digestive disorders [49]; a high abundance of *Flavonifractor* correlates with a low quality of life in large populations [50]; *Intestinimonas* is significantly reduced in obesity [51]; *Raoultella* shows a positive correlation with inflammatory bowel disease [52]; *Dorea*, a major gas-producing bacterium in the human gut, is prevalent in children with irritable bowel syndrome [53]; *Faecalibacterium* is ubiquitous in the gut of healthy individuals [54]; *Parabacteroides* has been positively correlated with anti-epileptic effects in gut microbiota studies [55]; *Holdemania* is significantly less abundant in the gut of kidney stone patients compared to healthy individuals [56]; *Streptococcus* exhibits a 30% higher variation in diarrhea populations than in asymptomatic populations [57]; *Erysipelatoclostridium* is significantly increased in kidney stone patients [58]; *Alistipes* is a dominant genus in the gut of patients with depression [59]. These findings suggest that consuming fresh radish sprouts and steamed radish sprouts for 30 s may help reduce the abundance of detrimental gut bacteria. *Lachnoclostridium*, *Phascolarctobacterium*, *Parasutterella*, and *Hungatella* exhibit relatively higher abundance in the CK group. *Parasutterella* is significantly reduced in the gut of kidney stone patients [59,60]; *Hungatella* is more abundant in the gut of Parkinson’s disease patients compared to healthy individuals [61]. *Megasphaera* (*Firmicutes*), *Bacillus* (*Firmicutes*), *Enterococcus* (*Firmicutes*), *Agathobacter*, *Comamonas* (*Proteobacteria*), *Megamonas* (*Firmicutes*), and two unidentified genera (unidentified_*Clostridiales* and unidentified_*Lachnospiraceae*) show the highest abundance in the P1 group; the remaining genera are most abundant in the FS group. *Megasphaera* is less abundant in the gut of patients with rheumatoid arthritis [62]; *Bacillus* is a normal, beneficial bacterial genus in the human gut [63]; *Enterococcus* is a normal gut flora, often residing in the gut and female urinary reproductive tract, and is an opportunistic pathogen [64]; *Megamonas* may be very abundant in the gut of Asian colorectal cancer patients [65], indicating that consuming steamed radish sprouts for 30 s is beneficial for maintaining normal gut flora.

### 3.3. The Impact of Radish Sprout Digesta Supplementation on Gut Microbiota Diversity—β-Diversity Analysis

Figure 4A illustrates the Principal Co-ordinates Analysis (PCoA) outcomes premised on Unweighted Unifrac distances. This analytical technique elucidates the principal constituents and structures within multidimensional datasets through the arrangement of eigenvalues and eigenvectors. The depiction delineates two principal components, which are responsible for explicating 14.93% and 11.09% of the dataset’s variance, respectively. Group P1 encompasses Group CK entirely, signifying an augmentation in microbial diversity consequent to the introduction of exogenous substances. Conversely, Group FS, while partially overlapping with Group CK, exhibits a considerably expanded area, denoting a pronounced enhancement in microbial diversity when compared to Group CK. The divergence in microbial diversity between Groups FS and P1 is ascribed to the dissimilarities in the composition of the additives. Collectively, the integration of both fresh and steam-processed radish sprout digestates enriches gut microbial diversity, with the steam-processed variant demonstrating a more pronounced impact.

Additionally, a UPGMA (Unweighted Pair Group Method with Arithmetic Mean) clustering analysis was conducted utilizing the Unweighted Unifrac distance matrix, and the clustering outcomes were amalgamated with the relative abundance of taxa at the phylum level for each sample, as depicted in Figure 4B. The underlying principle of UPGMA involves initially clustering the two samples with the shortest distance to form a new node, with the branch point positioned midway between the two samples. Subsequently, the average distance between this new “sample” and the remaining samples is determined, and the two samples with the smallest average distance are clustered together. This iterative process continues until all samples are integrated into a single cluster, culminating in a comprehensive dendrogram. The findings reveal that *Firmicutes*, *Bacteroidetes*, and *Proteobacteria* constitute over 80% of the microbial communities, aligning with prior research [66]. The dominance of *Firmicutes, Bacteroidetes,* and *Proteobacteria* in our study is consistent with their established roles in the gut ecosystem, and the observed changes in their proportions following the introduction of radish sprout digestates highlight the potential for dietary manipulation to modulate microbial balance [67]. The introduction of both digestates resulted in an increased diversity within *Firmicutes*, whereas the diversity of *Bacteroidetes* and *Proteobacteria* exhibited minimal variation between the CK and FS groups. In contrast to the CK group, the diversity of *Bacteroidetes* and *Proteobacteria* in the P1 group experienced a notable increase and decrease, respectively. *Actinobacteria* and *Fusobacteria* displayed negligible differences across the three groups.

To discern differential species across taxonomic hierarchies (Order, Family, Genus, Species) between groups, *t*-tests were employed to identify species with statistically significant differences (*p* < 0.05), as depicted in Figure 5. The FS and P1 groups exhibited a significantly higher abundance of *Bacillales* (Order), *Bacillaceae* (Family), and *Bacillus* (Genus) compared to the control group at their respective taxonomic levels. These taxa are classified under the *Bacilli* class of *Firmicutes* and are Gram-positive bacteria. Constituting a vital part of the gut microbiota, *Bacillus* species, including *Bacillus* subtilis (encompassing *Bacillus* natto), *Bacillus* cereus, and *Bacillus* licheniformis, are extensively utilized as probiotics in human applications [68,69]. Through their adhesive properties to the intestinal lining and antagonistic effects on pathogenic bacteria, *Bacillus* species contribute to the maintenance of gut microbiota equilibrium and intestinal health [70,71]. The incorporation of both fresh and steamed radish sprout digestates appears to be advantageous for gut microbiota balance and intestinal well-being. Moreover, at the species level, the FS group demonstrated a significantly higher abundance of *Clostridium-beijerinckii* compared to the control group. *Clostridium-beijerinckii*, a member of the *Clostridium* genus within *Firmicutes*, is recognized as a key butyrate-producing bacterium in the gut, aligning with our prior findings on short-chain fatty acids [72,73]. The findings suggest that the supplementation of fresh and steamed radish sprout digestates can enhance the abundance of *Bacillus*, which is instrumental in elevating the ratio of beneficial bacteria. Additionally, the enrichment of *Clostridium-beijerinckii* due to the addition of fresh radish sprout digestate implies a positive impact on butyrate synthesis.

## 4. Conclusions

In vitro fermentation of radish sprout digestates led to a significant increase in SCFAs such as acetic, propionic, and isobutyric acids within 48 h, with a right-skewed distribution. Valeric acid showed increased variability. The FeS group had lower propionic and isovaleric acids but higher butyric and valeric acids. *Phyla Firmicutes*, *Bacteroidetes*, *Proteobacteria*, *Fusobacteria*, and *Actinobacteria* constituted over 80% of the microbiota, with notable genera including *Bacteroides* and *Fusobacterium*. The introduction of radish sprout digestates increased *Firmicutes* diversity, with steamed sprouts (P1 group) enhancing diversity more effectively and reducing potentially harmful genera like *Bacteroides* and *Fusobacterium*, suggesting improved gut health benefits over fresh sprouts. *t*-test analysis showed higher abundances of *Bacillales*, *Bacillaceae*, and *Bacillus* in treated groups compared to the control. Future research should explore the long-term effects of such dietary interventions on gut microbiota composition and function, as well as the mechanisms underlying the observed changes in SCFA production. Additionally, the translational potential of these findings to human health and disease prevention warrants further investigation.

## Figures and Tables

**Figure 1 foods-14-00170-f001:**
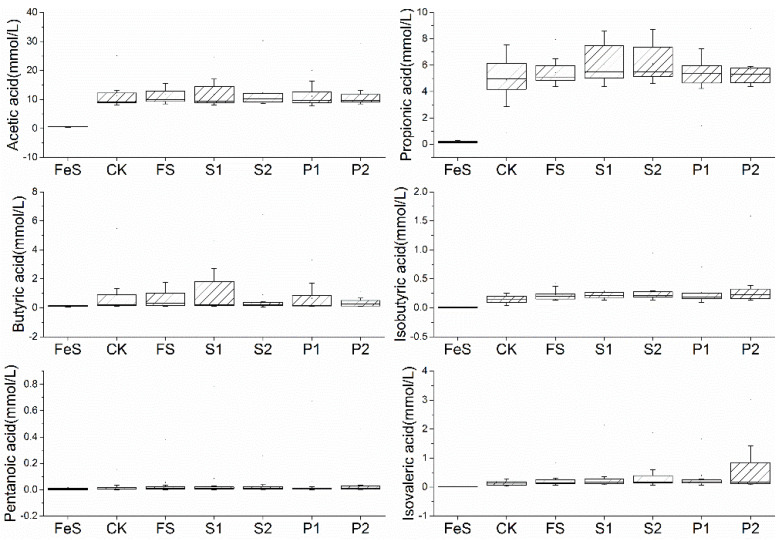
Effects of radish sprouts additives on short-chain fatty acids during the in vitro fermentation of intestinal bacteria in healthy individuals. FeS: fecal suspension. CK: control fermentation broth without intervention. FS: fermentation broth with fresh radish digest. S1: fermentation broth with radish sprout digest for 12 h storage at 25 ± 1 °C. S2 fermentation broth with radish sprout digest for 12 h storage at 4 ± 1 °C. P1: fermentation broth with radish sprout digest steamed 30 s. P2: fermentation broth with radish sprout digest steamed 300 s.

**Figure 2 foods-14-00170-f002:**
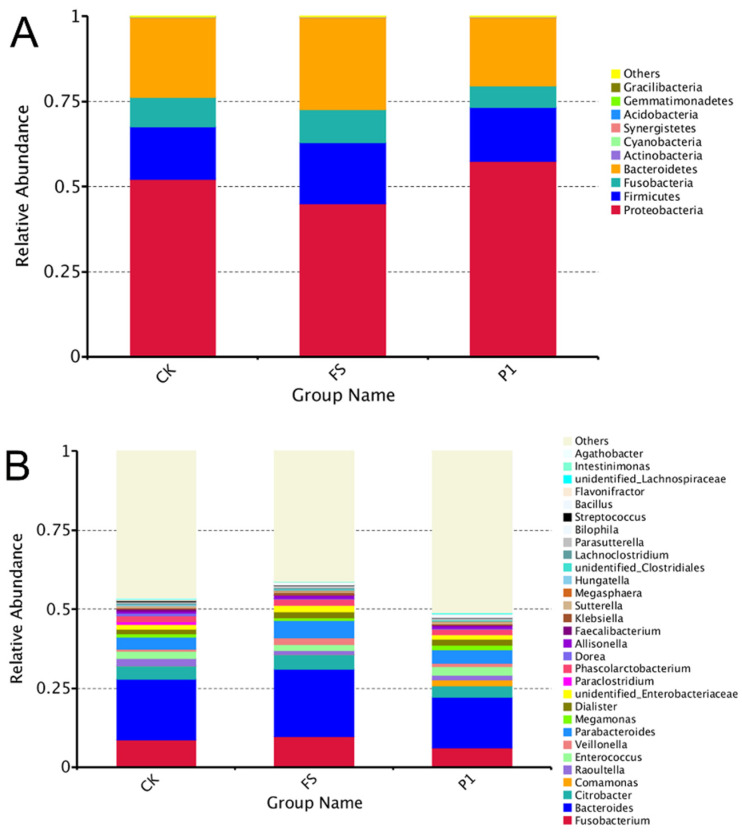
Species relative abundance of effects of radish sprouts additives on intestinal microflora in the in vitro fermentation broth. (**A**): At the phylum level; (**B**): at the genus level. CK: control fermentation broth without intervention; FS: fermentation broth with fresh radish digest; P1: fermentation broth with radish sprout digest steamed 30 s. Others: the sum of the relative abundances of all other species at the phylum (**A**)/genus (**B**) levels.

**Figure 3 foods-14-00170-f003:**
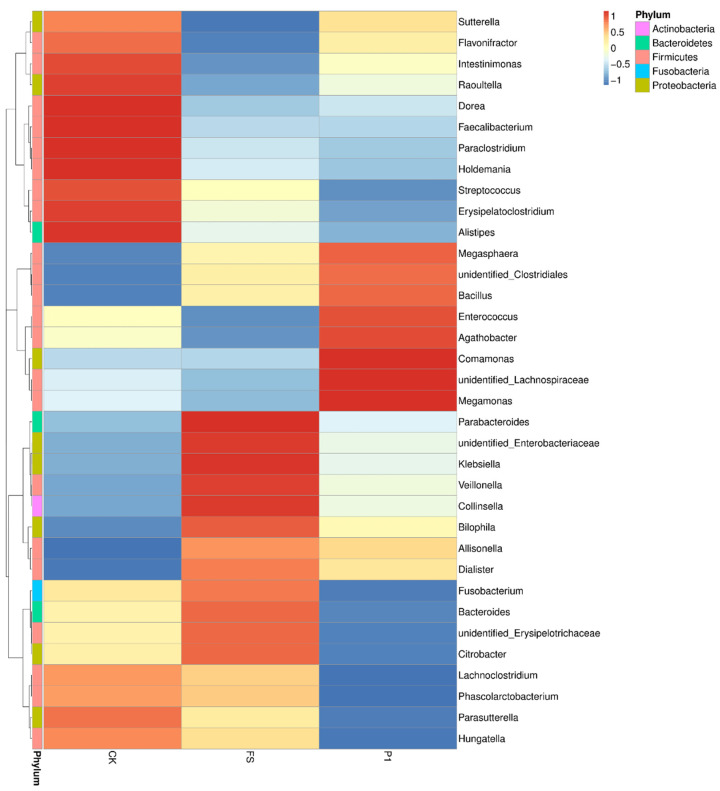
Cluster thermogram of radish sprouts additives on intestinal flora in the in vitro fermentation broth at the genus level. The longitudinal direction is the sample information, and the horizontal direction is the species annotation information. The cluster tree on the left side of the figure is the species clustering tree. CK: control fermentation broth without intervention; FS: fermentation broth with fresh radish digest; P1: fermentation broth with radish sprout digest steamed 30 s.

**Figure 4 foods-14-00170-f004:**
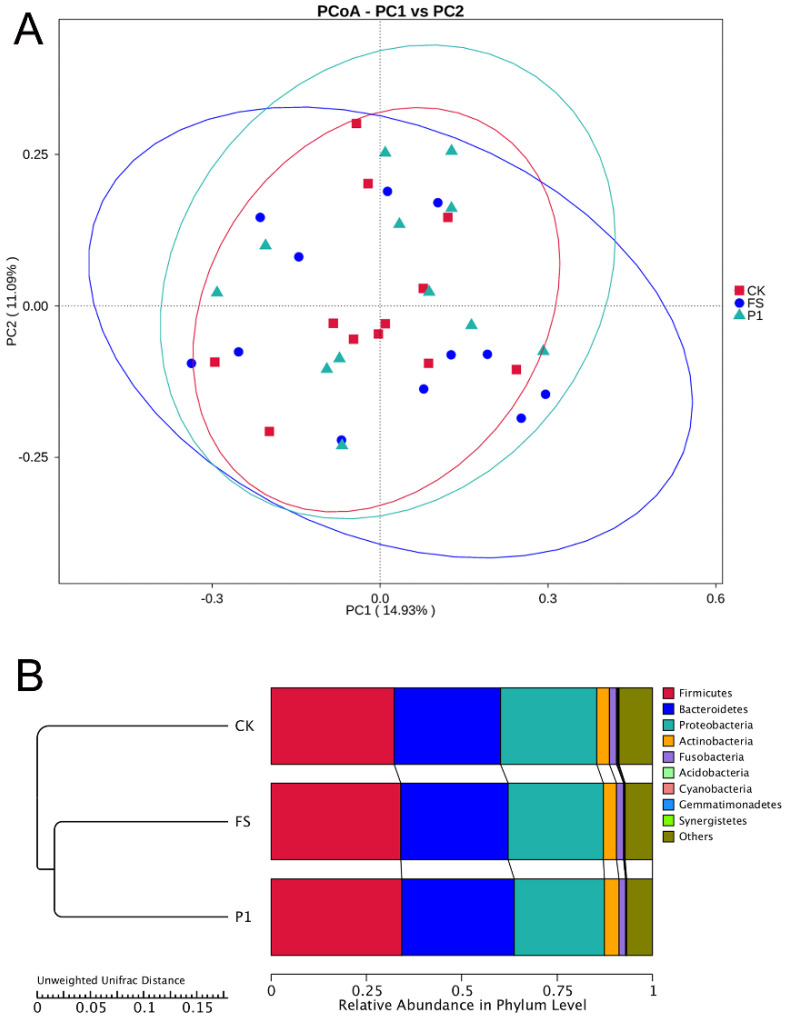
PCoA analysis (**A**) and UPGMA cluster tree (**B**) based on Unweighted Unifrac distance for the effect of radish sprouts additives on intestinal flora in fermentation broth. (**A**) The abscissa represents a principal component, the ordinate represents another principal component, and the percentage represents the contribution of the principal component to the sample difference; each point in the graph represents one sample, and the samples of the same group are represented by the same color. CK: control fermentation broth without intervention; FS: fermentation broth with fresh radish digest; P1: fermentation broth with radish sprout digest steamed 30 s. (**B**) On the left is the UPGMA clustering tree structure, and on the right is the relative abundance distribution of the species at the door level. CK: control fermentation broth without intervention; FS: fermentation broth with fresh radish digest; P1: fermentation broth with radish sprout digest steamed 30 s.

**Figure 5 foods-14-00170-f005:**
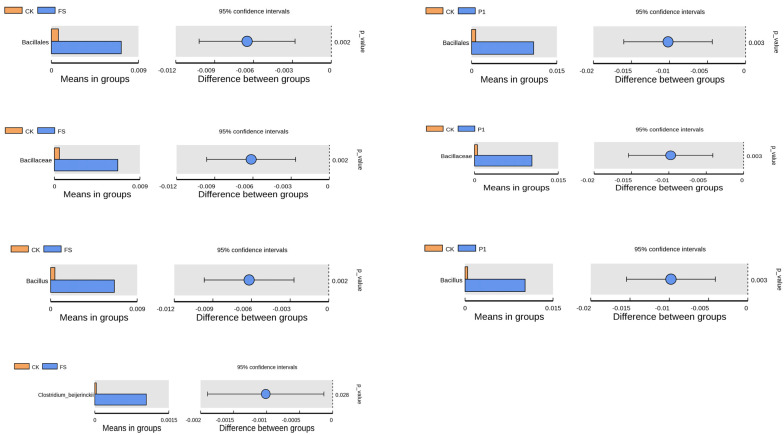
Species with significant differences between groups by *t*-test (from top to bottom: order, family, genus, and species) of the effect of radish sprouts additives on intestinal flora in the in vitro fermentation broth. CK: control fermentation broth without intervention; FS: fermentation broth with fresh radish digest; P1: fermentation broth with radish sprout digest steamed 30 s.

## Data Availability

The original contributions presented in this study are included in the article. Further inquiries can be directed to the corresponding author.

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
