# Peer review of "Study on the Effect of Radish Sprouts on Short-Chain Fatty Acids and Gut Microbial Diversity in Healthy Individuals"

_foods, 2025, doi:10.3390/foods14020170_

Round 1

Reviewer 1 Report

Comments and Suggestions for Authors

Title: Study on the Effect of Radish Sprouts on Short-Chain Fatty Ac- ids and Gut Microbial Diversity in Healthy Individuals

Foods-3376252

The manuscript presents valuable information on the crucial role of SCFAs in gut health and the influence of dietary interventions, specifically the beneficial effect of radish sprouts, on microbial composition. The study offers an interesting perspective on the relationship between radish sprouts and human gut health. While much remains to be understood regarding the interactions between dietary factors and their impact on gut health or microbial dysbiosis, this work represents a significant step toward advancing that understanding.

After a detailed review, I have identified a few minor items that should be addressed or corrected:

  • Bacterial names should be italicized. Please revise the entire text accordingly.
  • The quality of some figures could be improved.
  • The authors should follow the journal's guidelines more closely regarding the different sections. It may be helpful for the authors to consolidate the results and discussion into one section and provide a separate conclusions section.

Author Response

Comments 1: Bacterial names should be italicized. Please revise the entire text accordingly.

Response 1: Thank you for pointing this out. We agree with this comment. Therefore, We have modified the bacterial names in the manuscript to italics(in red).

Comments 2: The quality of some figures could be improved.

Response 2: Thank you very much for pointing this out. We have followed your suggestions and improved the quality of the figures in the manuscript. Please refer to Figures 1-5 in the manuscript.

Comments 3: The authors should follow the journal's guidelines more closely regarding the different sections. It may be helpful for the authors to consolidate the results and discussion into one section and provide a separate conclusions section.

Response 3: Thank you for your suggestion, we have combined the results and discussion sections (in red).

Reviewer 2 Report

Comments and Suggestions for Authors

The manuscript explores the in vitro and in vivo effects of radish sprouts additives on the gut microbiota. After carefully reading this work, I have the following observations:

1.      I suggest rereading the text for typos and grammatical mistakes.

2.      “in vitro” and “in vivo” should be italicized throughout the manuscript.

3.      Bacterial species names should be italicized throughout the manuscript as well.

4.      Line 49: Add the full name of the plant species used in the study with authorship at the first mention only, and then the abbreviation or the common name can be used afterward.

5.      Add a schematic diagram summarizing the experimental design.

6.      What was the selected cohort's average age and gender distribution?

7.      Add the details of the sequence trimming process.

8.      Lines 229-231, 258-272, 278-295, 320-346, 382-384, and 400-411: Avoid using references in the Results section. This section should focus on the findings of the current study. Relocate those parts to the discussion and rewrite.

9.      The discussion is too brief and lacks citations. This section should compare the main findings of this study to previous reports. I suggest merging the results and discussion to strengthen it.

10.  The conclusions are lengthy. Consider rewriting this section to provide a more concise summary of the key takeaways.

11.  Add the study’s approval number, Institutional Review Board Statement, and Informed Consent Statement. 

Author Response

Comments 1: I suggest rereading the text for typos and grammatical mistakes.

Response 1: Thank you for this suggestion. We have carefully reviewed the manuscript again and looked for typos and grammatical errors.

Comments 2: “in vitro” and “in vivo” should be italicized throughout the manuscript.

Response 2: Thank you very much for pointing this out. We have changed the format in the manuscript to italics.

Comments 3: Bacterial species names should be italicized throughout the manuscript as well.

Response 3: Thank you for pointing this out. We agree with this comment. Therefore, We have modified the bacterial names in the manuscript to italics.

Comments 4: Line 49: Add the full name of the plant species used in the study with authorship at the first mention only, and then the abbreviation or the common name can be used afterward.

Response 4: Thank you very much for your suggestion; we have added the full Latin name of radish sprouts in the manuscript.

Comments 5: Add a schematic diagram summarizing the experimental design.

Response 5: Thank you very much for your suggestion; we have followed your advice and created a schematic diagram of the experimental design, which can be referred to in the Graphical Abstract.

Comments 6: What was the selected cohort's average age and gender distribution?

Response 6: In this study, the volunteers were healthy Chinese individuals aged 20 to 40 years old, with a gender ratio of 1:5.

Comments 7: Add the details of the sequence trimming process.

Response 7: Raw data was extracted from the IonS5™ XL platform in fastq file format. Sample data was delineated based on barcode sequences, and chimeras are eliminated to ob-tain clean reads. Quality control metrics, including Raw Reads, Clean Reads, total Base count, Average Length, Q20 percentage, GC content, and Effective percentage, were assessed.

Comments 8: Lines 229-231, 258-272, 278-295, 320-346, 382-384, and 400-411: Avoid using references in the Results section. This section should focus on the findings of the current study. Relocate those parts to the discussion and rewrite.

Response 8: Thank you for your suggestion, we have combined the results and discussion sections.

Comments 9: The discussion is too brief and lacks citations. This section should compare the main findings of this study to previous reports. I suggest merging the results and discussion to strengthen it.

Response 9: Thank you for your suggestion, we have combined the results and discussion sections.

Comments 10: The conclusions are lengthy. Consider rewriting this section to provide a more concise summary of the key takeaways.

Response 10: Thank you for your suggestion, we have refined and condensed the conclusion section.

Comments 11: Add the study’s approval number, Institutional Review Board Statement, and Informed Consent Statement.

Response 11: Thank you for your advice; we have provided the Approval from the Ethics Committee of Human Research at China Agricultural University and the Subject Information Sheet and Informed Consent Form.

Reviewer 3 Report

Comments and Suggestions for Authors

The paper entitled „Study on the Effect of Radish Sprouts on Short-Chain Fatty Acids and Gut Microbial Diversity in Healthy Individuals” deals with the impact of stored and steamed radish sprout in vitro digestates on human gut microbiota. The topic is interesting and can be of relevance to the readers of Foods journal. However, there are some issues that should be solved in order to improve the quality of the presented research.

- Introduction – this part could be improved and expanded with more information on radish sprouts and their composition, with emphasis on the health beneficial compounds – for example phenolics.... Which phenolics present in radish sprouts are responsible for the suggested effect on microbiota? The benefits of sprouts consummation and their place in diet in general could be mentioned. Also, authors should point out the novelty of their study with reference to the similar ones.

- Materials and methods – gas chromatography method for short-chain fatty acid analysis should be adequately referenced. Namely, the reference 31 (in line 176) does not have GC method described, so the appropriate reference should be added.

- Results – the results are well presented with adequate figures and statistical tests. Figure 1 should be mentioned in the text (line 233). The explanation should be given how the groups CK, FS and P1 were chosen for gut microbiota analysis (Figure 2).

- Discussion - Discussion could be expanded with some literature data concerning previous research on the influence of radish sprouts on microbiome. Also, the parts from the Results section with references and comments should be transferred to Discussion (or Results and Discussion section should be merged).

Author Response

Comments 1: Introduction – this part could be improved and expanded with more information on radish sprouts and their composition, with emphasis on the health beneficial compounds – for example phenolics.... Which phenolics present in radish sprouts are responsible for the suggested effect on microbiota? The benefits of sprouts consummation and their place in diet in general could be mentioned. Also, authors should point out the novelty of their study with reference to the similar ones.

Response 1: Thank you for your suggestion; we have added the corresponding content to the introduction section of the manuscript.

Comments 2: Materials and methods – gas chromatography method for short-chain fatty acid analysis should be adequately referenced. Namely, the reference 31 (in line 176) does not have GC method described, so the appropriate reference should be added.

Response 2: Thank you for pointing that out; we have added the corresponding literature.

Comments 3: Results – the results are well presented with adequate figures and statistical tests. Figure 1 should be mentioned in the text (line 233). The explanation should be given how the groups CK, FS and P1 were chosen for gut microbiota analysis (Figure 2).

Response 3: Thank you very much for your suggestions; we have made the corresponding revisions in the manuscript.

Comments 4: Discussion - Discussion could be expanded with some literature data concerning previous research on the influence of radish sprouts on microbiome. Also, the parts from the Results section with references and comments should be transferred to Discussion (or Results and Discussion section should be merged).

Response 4: Thank you for your suggestion, we have combined the results and discussion sections.

Reviewer 4 Report

Comments and Suggestions for Authors

The manuscript "Study on the Effect of Radish Sprouts on Short-Chain Fatty Acids and Gut Microbial Diversity in Healthy Individuals" addresses a timely and significant topic in food science and human health. The study’s focus on radish sprouts as a functional food with potential benefits for gut microbiota diversity and SCFA production aligns well with current trends emphasizing gut health and personalized nutrition.

The experimental design is well-structured, utilizing 16S rDNA sequencing to assess microbial diversity, which provides a robust and reliable methodological framework. The comparison between fresh and steam-treated radish sprouts is particularly relevant, offering insights into how minimal processing techniques can enhance or preserve the functional properties of food.

I have reviewed the document and only found some inconsistencies from lines 138-141; please correct the chemical formulas 

-- The medium consists of (per 100 mL) 0.1 g of casein peptone, 0.25 g of yeast extract, 139 0.4 g of NaHCO3, 0.1 g of cysteine, 0.045 g of K2HPO4, 0.045 g of KH2PO4, 0.09 g of NaCl, 140 0.009 g of MgSO4·7H2O, 0.009 g of CaCl2--

Author Response

Comments 1: I have reviewed the document and only found some inconsistencies from lines 138-141; please correct the chemical formulas

-- The medium consists of (per 100 mL) 0.1 g of casein peptone, 0.25 g of yeast extract, 139 0.4 g of NaHCO3, 0.1 g of cysteine, 0.045 g of K2HPO4, 0.045 g of KH2PO4, 0.09 g of NaCl, 140 0.009 g of MgSO4·7H2O, 0.009 g of CaCl2—

Response 1: Thank you for pointing this out; we have changed the numbers in the chemical formulas in the manuscript to subscript format.

Round 2

Reviewer 2 Report

Comments and Suggestions for Authors

The authors responded to the suggested revisions and made significant changes in the revised version.